# Multidisciplinary Roles of LRRFIP1/GCF2 in Human Biological Systems and Diseases

**DOI:** 10.3390/cells8020108

**Published:** 2019-01-31

**Authors:** Masato Takimoto

**Affiliations:** Institute for Genetic Medicine, Hokkaido University, Hokkaido 060-0815, Japan; takimoto@igm.hokudai.ac.jp; Tel.: +81-11-706-5576

**Keywords:** LRRFIP1/GCF2, transcriptional repression, nucleic acid binding, cytoskeletal system, signal transduction, cancer

## Abstract

Leucine Rich Repeat of Flightless-1 Interacting Protein 1/GC-binding factor 2 (LRRFIP1/GCF2) cDNA was cloned for a transcriptional repressor GCF2, which bound sequence-specifically to a GC-rich element of epidermal growth factor receptor (EGFR) gene and repressed its promotor. LRRFIP1/GCF2 was also cloned as a double stranded RNA (dsRNA)-binding protein to trans-activation responsive region (TAR) RNA of Human Immunodeficiency Virus-1 (HIV-1), termed as TAR RNA interacting protein (TRIP), and as a binding protein to the Leucine Rich Repeat (LRR) of Flightless-1(Fli-1), termed as Flightless-1 LRR associated protein 1 (FLAP1) and LRR domain of Flightless-1 interacting Protein 1 (LRRFIP1). Subsequent functional studies have revealed that LRRFIP1/GCF2 played multiple roles in the regulation of diverse biological systems and processes, such as in immune response to microorganisms and auto-immunity, remodeling of cytoskeletal system, signal transduction pathways, and transcriptional regulations of genes. Dysregulations of LRRFIP1/GCF2 have been implicated in the causes of several experimental and clinico-pathological states and the responses to them, such as autoimmune diseases, excitotoxicity after stroke, thrombosis formation, inflammation and obesity, the wound healing process, and in cancers. LRRFIP1/GCF2 is a bioregulator in multidisciplinary systems of the human body and its dysregulation can cause diverse human diseases.

## 1. Introduction

LRRFIP1/GCF2 was initially discovered as a part of an artificial chimeric cDNA. After a bona fide LRRFIP1/GCF2 cDNA was cloned and shown to code a transcriptional repressor, diverse roles of this molecule in multiple biological processes, such as transcriptional regulation, signal transduction, and cytoskeletal remodeling, have been revealed. Dysregulations of LRRFIP1/GCF2 play broad and critical roles in the development of human diseases such as infections and autoimmune diseases, neurological, cardiovascular, metabolic diseases and cancer. Furthermore, dysregulations also impact human biological responses to disease, such as wound repair process. In cancer, LRRFIP1/GCF2 plays important roles in the development of malignant phenotypes, such as continuous growth, epithelial-mesenchymal transition (EMT), invasion/metastasis, escape from apoptosis, susceptibility, and resistance to anti-cancer drugs. As there were few molecules like LRRFIP1/GCF2, of which functions in cells are so broad and still being revealed in relevant to human biological systems and diseases, reviewing on this molecule is of current significance.

### Identification and Cloning of LRRFIP1/GCF2: Its Gene, mRNA/cDNA, and Protein

LRRFIP1/GCF2 was initially cloned as a N-terminal part of cDNA of GC-binding factor (GCF), which allegedly bound in a sequence-specific manner to a GC-rich element of the promotor of epidermal growth factor receptor (EGFR) gene as a transcriptional repressor [1]. However, Takimoto et al. revealed that GCF cDNA was an artificial chimeric molecule [2], in which a part of GCF cDNA was fused with an unrelated cDNA, and a full-length of bona fide cDNA for the repressor was cloned and termed as GCF2 by Reed et al. [3].

LRRFIP1/GCF2 was also cloned as a dsRNA-binding protein to the transactivation responsive region (TAR) RNA of HIV-1, termed as TRIP [4], and as a binding protein to the Leucine Rich Repeat (LRR) of Flightless-1 (Fli-1), a regulator of cytoskeleton, termed as Fli-1 leucine rich repeat associated protein 1 (FLAP1) and LRRFIP1 [5,6]. TRIP, FLAP, and LRRFIP1 are identical to GCF2. In this review, this molecule is referred to as LRRFIP1/GCF2.

Genomic DNA size of LRRFIP1/GCF2 is about 73 kbp and consists of 29 exons [7]. Most human tissue and cancer cell lines express 4.2 kb mRNA and additional transcripts with 6.6, 2.9 and 2.4 kb have been found in several types of tissue [7]. Initially, a part of cDNA was cloned in the N-terminal of an artificial fusion molecule GCF cDNA [1,2]. Reed et al. had cloned a full-length cDNA clone, which had an open reading frame of 2256 nucleotides. The encoded protein had transcriptional repressive activity with a sequence-specific DNA-binding ability to the GC-rich sequence [3]. To date, five different isoforms of human LRRFIP1/GCF2 mRNA/cDNA have been identified and they encode proteins with amino acids ranging from 349 to 808 in number [8]. The endogenous LRRFIP1/GCF2 proteins in several human cell lines, the one expressed from LRRFIP1/GCF2 cDNA in reticulolysates, and the recombinant LRRFIP1/GCF2 protein produced in bacteria have migrated as a band with a molecular weight (MW) of 160 kda in sodium dodecyl sulfate-polyacrylamide gel electrophoresis (SDS-PAGE). Mass spectroscopic analysis of the recombinant protein, however, determined the MW of LRRFIP1/GCF2 proteins as 83 kDa, which conformed to the one calculated from the amino acid component of this protein. This discrepancy could be due to the high content of acidic amino acids and the presence of a highly basic region of this protein, and the charge interaction between the acidic and basic regions [3,7].

LRRFIP1/GCF2 protein consists of three domains (Figure 1); an N-terminal Helix domain, a central coiled-coil domain and a C-terminal Nucleic acid binding domain. The coiled-coil domain is necessary for the interactions with Fli-1 and for self-dimerization [6,8]. The Nucleic acid binding domain binds to DNAs and RNAs, and the middle of this domain, which is called as Lysine rich motif, is crucial for sequence-specific binding to the double-stranded GC-rich DNA (dsDNA) element and for transcriptional repression [7]. Moreover, it is and required for LRRFIP1/GCF2 protein to bind to dsRNA, TRIP [4].

Yang et al. showed that an N-terminal region of LRRFIP1/GCF2 protein, which was not delineated in their study, was indispensable for binding to dsDNA [9]. Arakawa suggested that the region that spans N-terminal Helix, coiled-coil domains and a part of Nucleic acid binding domain have weak homology to the sequence motifs for binding to dsRNA [10]. Therefore, it is probable that, in addition to the C-terminal Nucleic acid binding domain, the N-terminal and the central domains may play some role in binding LRRFIP1/GCF2 protein to nucleic acids.

LRRFIP1/GCF2 binds to β-catenin, glucocorticoid receptor interacting protein 1 (GRIP1), and p300 [9,11]. The binding domains for them, however, have not been identified. The protein domain structure of LRRFIP1/GCF2 in relation to its molecular interactions and functions is depicted in Figure 1.

## 2. Biological Functions of LRRFIP1/GCF2

LRRFIP1/GCF2 protein plays multiple roles in diverse cellular processes in the nucleus and cytoplasm.

### 2.1. LRRFIP1/GCF2 as a Transcriptional Repressor

LRRFIP1/GCF2 protein was initially characterized as a transcriptional repressor with a sequence specific DNA-binding activity to an upstream element of epidermal growth factor receptor (EGFR) gene. Reed et al. have shown that LRRFIP1/GCF2 protein is bound to a region located between −249 and −233 bps from transcriptional start site (TSS) of EGFR gene and repressed its transcription [3]. Potential binding sites for LRRFIP1/GCF2 have also been observed in other growth factor/receptor genes, such as insulin like growth factor-II (IGF-II) and tumor growth factor-α(TGF-α) [17,18]. Down regulation of EGFR by nerve growth factor (NGF) has been shown to be accompanied by an increase of cellular content of LRRFIP1/GCF2, suggesting that LRRFIP1/GCF2 plays a role in inhibiting the overgrowth of cells [19].

Subsequent studies have revealed that the promoters of platelet derived growth factor-A (PDGF-A) gene were also bound and repressed by LRRFIP1/GCF2 [20]. The variant TNF-α gene promotor, which is not bound nor repressed by LRRFIP1/GCF2 is discussed in relation to autoimmune diseases in Section 3.1 [21]. It is interesting and noteworthy that the binding site for LRRFIP1/GCF2 in the promotor of PDGF-A gene overlapped with those for transcriptional activators, Sp1, Egr-1 [20]. Repression of transcription by LRRFIP1/GCF2 has also been suggested in regard to Glutamate transporter EAAT2 gene [22], which is referred to in Section 3.2.

A recent report revealed that non-coding RNAs transcribed in the upstream of TSS of TNF-α gene play important roles in the repression of TNF-α promotor by binding to LRRFIP1/GCF2 protein, and that the binding of LRRFIP1/GCF2 to genomic DNA required other repressor proteins, EHZ2 and SUZ12 [23,24,25]. The report also showed that the binding affinities of LRRFIP1/GCF2 protein to single and dsRNAs were higher than to that of dsDNA [23]. This result conformed to the study by Wilson et al., which showed that LRRFIP1/GCF2 protein was identified as a TAR RNA interacting protein, TRIP, which bound to the dsRNA of HIV-1 [4].

### 2.2. LRRFIP1/GCF2 as a Regulator for Cytoskeletal System

Interestingly, LRRFIP1/GCF2 was also identified as an interacting protein to Fli-1, which has a LRR and a gelsolin-like domain [5,6]. As gelsolin is a cytoskeletal remodeling factor by acting on actin [26], this observation suggests that LRRFIP1/GCF2 plays a role in cell morphology and reorganization of cytoskeletal system, as Fli-1 does in early embryonic development in Fruit fly and mammals.

When cells adhere to fibronectin, integrin receptor-dependent signaling pathways are activated in the cells and this induces the remodeling of their cytoskeletal system that leads to changes of migration capacity in the tissue. RhoA is a small GTPase and a pivotal molecule in the regulation of the cytoskeletal system, working downstream of the fibronectin induced-integrin pathway. Inhibitions of LRRFIP1/GCF2 by the transfection of short inhibitory RNA (siRNA) into a cervical carcinoma and colorectal cancer cell lines have been shown to reduce RhoA activation, migration, and cytoskeletal remodeling of the transfected cell lines [13,15]. These studies suggested the positive regulatory role of LRRFIP1/GCF2 in RhoA activation. Ariake et al. also showed that Leukemia-associated Rho-specific guanine nucleotide exchanging factor (LARG), which is an activator for RhoA, was bound through its Regulator of G-protein Signaling (RSG) domain by LRRFIP1/GCF2, suggesting that LRRFIP1/GCF2 activated RhoA through its interaction with LARG [15,27,28]. One report, however, showed that over-expression of LRRFIP1/GCF2 repressed the expression of RhoA and disorganized the cytoskeletal system in an epidermoid carcinoma cell line [29]. Further studies are required to address the discrepancy regarding the role of LRRFIP1/GCF2 in the regulation of RhoA.

The role of LRRFIP1/GCF2 in the activation of RhoA is also described in the following section in regard to the role of LRRFIP1/GCF2 as a regulator in signal transduction.

### 2.3. LRRFIP1/GCF2 as a Regulator in Signal Transduction

LRRFIP1/GCF2 protein works as a co-stimulator for signals from the cell surface, triggered by molecules such as Wnt, hormones, and extracellular matrix. In the Wnt/canonical pathway which is dependent on β-catenin, Wnt binds its cell surface receptor, Frizzled, and the signal is relayed to Dsv [13,30]. LRRFIP1/GCF2 interacts with Dsv and β-catenin, and the signal is further transduced to the nucleus. LRRFIP1/GCF2 enhances the interactions of β-catenin with transcriptional activators, such as lymphocyte enhancer binding factor 1/T cell factor (LEF1/TCF), and with a coactivator creb-binding protein (CBP)/p300 in the nucleus. LRRFIP1/GCF2 is strongly synergistic with CBP/p300, and this functional synergy is disrupted by Fli-1, indicating that LRRFIP1/GCF2 stimulates the Wnt/canonical signaling pathway, while Fli-1 counteracts and inhibits this β-catenin dependent pathway [16,30].

Labbe et al. identified several isoforms of LRRFIP1/GCF2 and showed that one isoform had the highest enhancing activity on the Wnt/β-catenin pathway, and that two tryptophan residues in an N-terminal domain in the isoform played a critical role in the activation [8]. LRRFIP2, a homologous protein of LRRFIP1/GCF2 and a binding protein to LRR of Fli-1, was also shown to bind to Dsv and activate the Wnt/β-catenin pathway [6,31].

In addition to the β-catenin dependent Wnt/canonical signal transduction pathway as described above [30], LRRFIP1/GCF2 also plays a role in β-catenin independent signaling, the Wnt/noncanonical pathway [32]. Ohtsuka et al. showed that LRRFIP1/GCF2 bound to Dsv, which is a mediator for both the Wnt/canonical and Wnt/ noncanonical pathways, and that siRNA knockdown of LRRFIP1/GCF2 inhibited the activation of RhoA by Wnt treatment in human cancer cell lines. As RhoA is a target molecule of the β-catenin independent pathway, the results suggested the essential role of LRRFIP1/GCF2 in the Wnt/noncanonical pathway [13].

Ariake et al. showed that LRRFIP1/GCF2 played a role in regulating the integrin signaling pathway, which is stimulated by an extracellular matrix molecule, fibronectin, and the pathway led to RhoA activation. Knockdown of LRRFIP1/GCF2 reduced the activation of RhoA in the cells plated on fibronectin-coated dishes. By binding to LARG, LRRFIP1/GCF2 mediated the integrin signaling pathway and caused RhoA activation, working as a co-factor of LARG for RhoA activation [15].

LRRFIP1/GCF2 also plays roles in the nuclear receptor (NR) dependent pathway. It was shown that LRRFIP1/GCF2 bound to a co-activator glucocorticoid receptor interacting protein 1, GRIP1 [16], and that two LRRFIP1/GCF2 interacting proteins, β-catenin and Fli-1, participated in an NR dependent pathway; β-catenin binds to androgen receptor and activates androgen-stimulated transcription [33,34], and Fli-1 is a coactivator for NRs dependent gene activation [11]. Therefore, it is likely that LRRFIP1/GCF2 plays a role in hormone dependent cellular responses by regulating the gene expressions through NRs.

The mechanism of the biological functions of LRRFIP1/GCF2 is summarized in Figure 2.

## 3. Implications for Human Disease

### 3.1. Roles of LRRFIP1/GCF2 in the Defense against Viral and Bacterial Infections and in Regulation of Autoimmune Disorders

LRRFIP1/GCF2 protein plays a role as a cytosolic sensor for nucleic acids, which are exogenous and endogenous cellular DNAs and RNAs from microbial infection and apoptotic and necrotic dead cells. The binding of LRRFIP1/GCF2 to microbial nucleic acids, such as dsDNA derived from Listeria monocytogenes, negative-sense single-stranded RNA (sRNA) from Vesicular stomatitis virus (VSV), and positive-sense sRNA from Hepatitis C Virus (HCV), trigger β-catenin and Toll-like receptor (TLR) signaling pathways [9,10,35,36,37].

The former pathway leads to the phosphorylation of β-catenin at Ser552 potentially by Akt (Protein kinase B), the dissociation from cadherin at the plasma membrane and the increased cytosolic and nuclear translocation of β-catenin [35,38]. In the nucleus, LRRFIP1/GCF2 enhances the interactions of β-catenin with a transcription activator, Interferon regulatory factor 3 (IRF3), and a coactivator CBP/p300, increasing the transcriptions of type 1 interferon (IFN) gene, such as Interferon β (IFN-β). The latter pathway leads to the NF-κB activation by two mechanisms: The first; the binding of LRRFIP1/GCF2 to myeloid differentiation factor 88 (MyD88), an intracellular adaptor protein downstream of TLR, is increased in response to nucleic acids or LPS. The second; LRRFIP1/GCF2 competes with FliiH, a Fli-1 homologue and a negative regulator of the TLR pathway, for the binding to MyD88 and augments of NF-κB activity. LRRFIP2, a LRRFIP1/GCF2 homologue, also competes with FliiH to interact with MyD88 for the activation of NF-κB [14,39,40]. Thus, the binding of LRRFIP1/GCF2 to nucleic acids induces the production of type 1 IFN and pro-inflammatory cytokines [40,41]. It is reasonable to speculate that dysfunction of LRRFIP1/GCF2 would lead to vulnerability to infectious disease.

Duan et al. reported an interesting observation that β-catenin directory binds to and inhibits NF-κB, negatively regulating inflammation in non-virulent salmonella infection. This observation suggests that β-catenin plays a pivotal role in selecting the responses, whether to produce type 1 IFN or pro-inflammatory cytokines depending on the degree of the virulence of infected bacteria [42,43].

Various autoimmune diseases have been reported to associate with polymorphism of TNF-α gene promotor sequence [44]. LRRFIP1/GCF2 appears as a repressor to the promotor of TNF-α gene, presumably binding to G form of -308 site of the promotor in the cells that do not express TNF-α, while Ets-1, a transcriptional activator, probably binds to A form of the site in the cells that express TNF-α. The decreased binding of LRRFIP1/GCF2 to TNF-α promotor is likely to cause over-expression of TNF-α and various autoimmune diseases, such as rheumatoid arthritis [21].

### 3.2. Role of LRRFIP1/GCF2 in Neurological Disorders

Glutamate transporter excitatory amino acid transporter 2 (EAAT2, GLT-1) gene has polymorphism in its promotor sequence, in which wild-type promotor has a binding site for activator protein-2 (AP2), while a variant one has a site for LRRFIP1/GCF2. The variant is probably associated with a decreased level of EAAT2 protein, elevated glutamate level in plasma and synapses, and with a higher frequency of early neurological worsening in stroke of the variant patients [22], indicating that LRRFIP1/GCF2 represses the expression of EAAT2 in the variant, and then the elevated glutamate contributes to excitotoxity after a stroke.

Cerebral ischemia causes the death of nerve and glial cells which release proteins, glycans, and nucleic acids. Some of these molecules, including DNAs and RNAs, activate β-catenin and TLRs signaling pathways as ligands to LRRFIP1/GCF2 and TLRs. The activated TLRs pathways induce through NF-κB pro-inflammatory cytokines, which cause worsening of brain stroke [45]. Treatments with antagonists to TLRs, such as TAK424 or MicroRNA-1906, were shown to protect brain ischemia [46,47]. Interestingly, pre-treatments with TLR agonist, such as LPS, also protected cerebral ischemia by induction of the inhibitors of inflammation, a phenomenon called as ischemic tolerance or preconditioning [45,48,49,50]. These results indicate that there is an association between immune and neurological systems through LRRFIP1/GCF2, β-catenin and TLRs.

An isoform of rat LRRFIP1/GCF2 was significantly induced in an ischemic rat brain, especially in the peripheral area of infarction after middle cerebral artery occlusion. siRNA knockdown of LRRFIP1/GCF2 in cultured rat astrocytes reduced the levels of phosphorylated forms of β-catenin, Akt and mammalian target of rapamycin (mTOR) [51]. As the phosphorylated forms of β-catenin and Akt/mTOR stimulate neurogenesis and inhibit apoptosis [52], they are suggested to work in neuro-protective ways after oxygen glucose deprivation and injury [52,53,54,55,56]. In addition, β-catenin was reported to inhibit NF-kB activation [42,43], by which LRRFIP1/GCF2 could protect the stroked brain from exacerbation. This is another example of the association between immune and neurological systems. Although LRRFIP1/GCF2 repressed the transcription of EAAT2 gene in its variant promotor [22], Gubern et al. reported that LRRFIP1/GCF2 increased EAAT2 protein level in cerebral cells, probably by mechanisms of post-transcriptional initiation through the PI3K/Akt/mTOR pathway [51,57]. The increase in EAAT2 protein, which could prevent excitotoxicity, may be beneficial to the brain after a stroke. The induced expression of LRRFIP1/GCF2 must be playing a positive role in the recovery of brain from ischemia by mediating these molecules.

Molecular responses to exogenous and endogenous nucleic acids, regulated by LRRFIP1/GCF2, are depicted in Figure 3.

### 3.3. Role of LRRFIP1/GCF2 in Cardiovascular Disease

A proteomic study on human platelets revealed that LRRFIP1/GCF2 functioned as a regulator for platelet cytoskeleton, interacting with Fli-1 and an isoform of Drebrin (DBN1), both of which play a role in cytoskeletal remodeling of platelets. Inhibitions of LRRFIP1/GCF2 by antisense Morpholino and by short inhibitory RNA (shRNA) expressed from lentivirus reduced the artificially induced-thrombosis formations in Zebrafish and mice, respectively, suggesting the potential use of them in prevention and therapy of deep vein thrombosis and myocardial infarction [58,59].

Khachigian et al. reported that LRRFIP1/GCF2 was induced by balloon injury of a rat carotid artery wall and that transfection of LRRFIP1/GCF2 cDNA into a vascular smooth muscle cell (VSMC) line reduced the proliferation of the cell line within 24 h after the transfection [20]. Choe et al. showed that the cultured cells transfected with LRRFIP1/GCF2 cDNA decreased in number at day 1 but increased at 3 and 5 days after the transfection [60]. They also showed that the injury of a rat carotid artery induced a microRNA, miR-132, which is supposed to target to LRRFIP1/GCF2, and that transfection of miR-132 mimic significantly inhibited the proliferation of VSMC and blocked neointimal hyperplasia after the carotid injury, indicating the potential use of miR-132 in the prevention of the neointimal hyperplasia after balloon injury and of atherosclerosis.

It was reported that TLRs modulated the responses to myocardial ischemia and infarction [61,62]. There is, however, no published research that suggests the direct involvement of LRRFIP1/GCF2 in those responses. 

### 3.4. Roles of LRRFIP1/GCF2 in Metabolic Disease

Several single nucleotide polymorphism (SNP) variants of LRRFIP1/GCF2 have been shown to be associated with phenotypes for obesity and inflammation. One of the SNS variants has the strongest correlation with adiposity-phenotypes, such as visceral and subcutaneous adipose tissue areas, and plasma level of C-reactive protein (CRP), a marker protein for inflammation [63].

There were studies that indicated the association of TLR with body adiposity and obesity; a TLR- 4 variant was associated with adiposity of the body and liver [64], and TLR-4 deficiency protected obesity induced by diets with high saturated fat [65]. As described above in Section 3.1, LRRFIP1/GCF2 protein regulates TLR signaling, suggesting the role of LRRFIP1/GCF2 in inflammation [10,40]. Those studies may provide the molecular basis for the links and relations among LRRFIP1/GCF2, inflammation, and obesity.

### 3.5. Roles of LRRFIP1/GCF2 in the Wound Healing Process

A recent study revealed that LRRFIP1/GCF2 plays an important role in the repair processes of acute wounds [66]. In the response to incisional skin wounding in mice, LRRFIP1/GCF2 was significantly up-regulated in fibroblasts and keratinocytes in the dermis and epidermis of the skin. Treatment with recombinant LRRFIP1/GCF2 protein (rLRRFIP1/GCF2) increased cell proliferation of cultured fibrocytes and keratinocytes in vitro and stimulated wound repair process and cell proliferation in vivo. The rLRRFIP1/GCF2 treatment also decreased the level of Fli-1, which is a negative regulator for wound healing [67,68], and increased total collagen synthesis. In wounded skin legions, TLR-4 mediated inflammation was inhibited by the rLRRFIP1/GCF2 treatment, manifested by the decrease in number of infiltrating neutrophils and macrophages. The rLRRFIP1/GCF2 decreased TGF-β1 expression and increased TGF-β3 which play roles in pro-scarring and anti-scarring, respectively [69]. These results suggest that LRRFIP1/GCF2 plays a role in the anti-scarring process.

The biological functions of LRRFIP1/GCF2 and human diseases are summarized in Table 1.

### 3.6. Roles of LRRFIP1/GCF2 in Development of Cancer and Malignant Phenotypes

In a patient with hematological malignancy of 8p11-chromosome eight-myeloproliferative syndrome (EMS), in which fibroblast growth factor receptor 1 (FGFR1) gene was involved in chromosomal translocations, LRRFIP1/GCF2 gene was identified as a new partner gene for FGFR1. In this translocation, LRRFIP1/GCF2 gene on chromosome 2p37 and FGFR1 gene on chromosome 8p11 was reciprocally translocated, producing an in-frame fusion transcript of LRRFIP1/GCF2 exon 9 with FGFR1 exon 9, as well as an in-frame fusion transcript of FGFR1 exon 8 with LRRFIP1/GCF2 exon 10 [70]. The first in-frame fusion transcript encoded a predicted chimeric protein with 668 amino acids, which had the coiled-coil region of LRRFIP1/GCF2 and two tyrosine kinase domains of FGFR1. The coiled-coil region could have contributed to dimerization of the tyrosine kinase domains, leading to the constitutive activation of the kinase domain, which has been revealed in earlier cases of FGFR1 rearrangements in EMS.

As the protein encoded by one isoform of LRRFIP1/GCF2 was highly expressed in a human hepatocellular carcinoma (HCC) cell line, HepG2, and in most primary HCC tissues, LRRFIP1/GCF2 was identified as a tumor associated antigen of HCC, which could be a potential valuable biomarker for early diagnosis of HCC [71]. Knockdown of LRRFIP1/GCF2 by RNAi caused cell growth inhibition and the increased apoptosis of the HepG2 cell line [72]. Reduced expression of LRRFIP1/GCF2 in HepG2 cell restored the activity of caspase 3, which is the potent executer for programmed cell death. These results indicate that over-expression of LRRFIP1/GCF2 is favorable for cancer cells to proliferate and escape from apoptosis.

LRRFIP1/GCF2 was shown to promote EMT in pancreatic cancer through the Wnt/β-catenin pathway. Knockdown of LRRFIP1/GCF2 reversed EMT, with the increased expression of E-cadherin, an epithelial marker, and the decreased expression of vimentin, a mesenchymal marker, and the migration and invasion capacities of pancreatic and lung cancers being significantly inhibited [73]. EMT, which is a crucial mechanism that induces tumor metastasis and invasion, was stimulated by LRRFIP1/GCF2 through the Wnt/β-catenin pathway.

LRRFIP1/GCF2 promoted metastasis and liver invasion of human colorectal carcinoma cell lines transplanted in mice through RhoA activation. RhoA plays a role in regulating downstream in the integrin signaling pathway, and knockdown of LRRFIP1/GCF2 reduced activation of RhoA in the colorectal cancer cells plated on fibronectin-coated dishes. LARG binds to LRRFIP1/GCF2 and mediates the integrin signaling pathway, causing RhoA activation, leading to the stimulation of cytoskeletal remodeling, increased migration, and invasion of metastatic colorectal cancer cells (Figure 2). LRRFIP1/GCF2 shRNAs suppressed metastasis of the xenograft, accompanied by reduced adhesion of the cell lines to extracellular matrix, such as fibronectin. Migration capacity of the LRRFIP1/GCF2 siRNA-transfected cells was also compromised, probably by reduced expression of RhoA [15].

In glioblastoma multiforme (GBM), a human malignant brain tumor, the resistance to a topoisomerase II inhibitor teniposide (VM-26) was mediated by the over-expression of a microRNA, miR-21, which binds in the 3′-untranslated region (UTR) of LRRFIP1/GCF2 and down-regulates its expression [74]. The introduction of miR-21 anti-sense oligonucleotide into a GBM cell line up-regulated the expression of LRRFIP1/GCF2 and reduced the survival of the transfected cells treated with VM-26. The patients with a higher expression of LRRFIP1/GCF2 had a better prognosis with VM-26 treatment, and the tumor volumes of GBM cells, which were transfected with a RRFIP1/GCF2-expressing plasmid and then transplanted in nude mice, saw an increased reduction with VM-26 treatment compared with a non-transfected control. These results suggest a potential role of LRRFIP1/GCF2 in chemo-sensitivity of GBM to VM-26 [75].

The epidermoid carcinoma cell lines with over-expressed LRRFIP1/GCF2 were reported to be sensitive to doxorubicin and vincristine, but resistant to cisplatin. In these cell lines with high LRRFIP1/GCF2 expression, the down-regulated of RhoA, and the disrupted actin-filament network were observed, and then the membrane-transporter multidrug resistance protein 1 (MRP1), which facilitates efflux of anti-cancer drugs from cytoplasm, was internalized from membrane to cytoplasm, leading to inefficient efflux and increased accumulation of doxorubicin and vincristine, and subsequent therapeutic effects. Resistance to cisplatin in the cell lines with the high LRRFIP1/GCF2 expression was at least due to significant reduction of cisplatin accumulation in cytoplasm, but it has not yet been fully elucidated why the cells accumulated less cisplatin [29].

The role of LRRFIP1/GCF2 in cancers are summarized in Figure 4.

As LRRFIP1/GCF2 was initially identified as a transcriptional repressor for several growth factors and their receptors, it was presumed that it played a role in inhibiting tumor growth. It, however, turned out that LRRFIP1/GCF2 worked rather advantageously for cancer by stimulating growth and inhibiting apoptosis. One exception was that LRRFIP1/GCF2 appears to function to inhibit the overgrowth of cells treated with nerve growth factor as a player in the negative feedback mechanism [19]. Another exception was that chemo-sensitivities to anti-cancer drugs, which accompanies growth effects, were affected with the level of LRRFIP1/GCF2 expression, in which some drugs worked in a way that inhibited growth in the presence of high LRRFIP1/GCF2 expression, while some induced resistance. LRRFIP1/GCF2 expression also contributed to malignant phenotypes of cancerous cells, such as promoting EMT, invasion, and metastasis. This could be mainly attributed to the dysregulated roles of LRRFIP1/GCF2 in signal transduction pathways and in the cytoskeletal system.

## 4. Conclusions

LRRFIP1/GCF2 is a rare molecule that plays diverse functional roles, which span several pathophysiological systems and processes, such as immune, neurological, cardiovascular, metabolic, and wounds healing processes. This may be due to the fact that the human genome has fewer genes than initially expected. By allotting more than a single functional role to one protein with help from alternative splicing and having more than one functional domain, the human being was able to reduce the size of its genome, which otherwise would require a tremendous number for such a higher animal. Small genome size is beneficial for efficient replication, the maintenance of biological integrity, and energy saving. The multidisciplinary functions of LRRFIP1/GCF may also play an integrative role in regulating the whole human body. For these reasons, LRRFIP1/GCF2 is a representative and enticing molecule.

## Figures and Tables

**Figure 1 cells-08-00108-f001:**
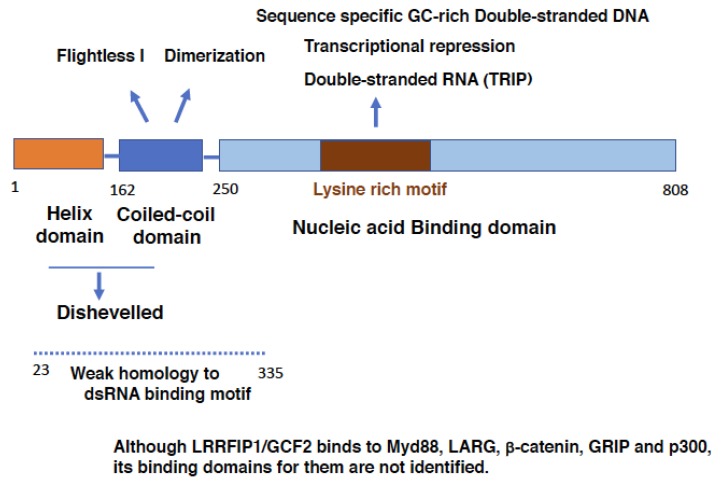
**Domain structure of Leucine Rich Repeat of Flightless-1 Interacting Protein 1/GC-binding factor 2 (LRRFIP1/GCF2) protein.** LRRFIP1/GCF2 protein consists of three domains; an N-terminal Helix domain [12], a central coiled-coil domain and a C-terminal Nucleic acid binding domain. The coiled-coil domain is necessary to interact with Flightless I and for self-dimerization [6,8]. The C-terminal Nucleic acid binding domain binds to DNAs and RNAs, and the middle of this domain, Lysine rich motif, is critical for sequence-specific binding to the double-stranded GC-rich DNA element, for transcriptional repression and for binding to dsRNA TRIP [4]. The dotted line is the region that has weak homology to the sequence motifs for binding to dsRNA [10]. The N-terminal domain, which was not delineated, has also been suggested as playing a role in binding to dsDNA [9]. As LRRFIP1/GCF2 protein has five isoforms, a representative isoform, isoform 3 with 808 amino acids, is depicted in this figure [8]. As the binding domains for Disheveled (Dsv) and the sequence specific DNA-binding and the critical domain for the transcriptional repression were studied in isoform 5 [7,13], the corresponding domains were surmised in this figure. LRRFIP1/GCF2 binds to MyD88, (LARG), β-catenin, GRIP1, and p300 [9,14,15,16]. The binding domains for them, however, have not been identified. Abbreviations used; LRRFIP1/GCF2: Leucine Rich Repeat of Flightless-1 Interacting Protein 1/GC-binding factor 2, TRIP: TAR RNA interacting protein, Myd88: Myeloid differentiation factor 88, LARG: Leukemia associated Rho-specific guanine nucleotide exchange factor, GRIP: Glucocorticoid receptor interacting protein.

**Figure 2 cells-08-00108-f002:**
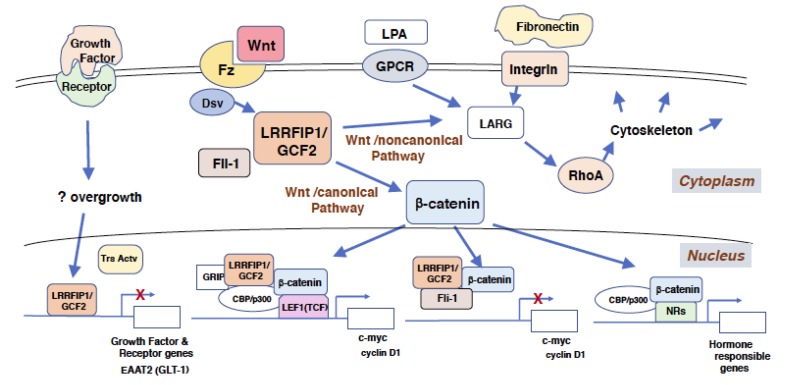
**Roles of LRRFIP1/GCF2 in transcriptional repression, signal transductions, and regulation on the cytoskeletal system.** Signals from outside of the cell, such as Wnt and extracellular matrix, are transduced to nucleus for gene transcription and to cytoplasm and cell membrane for cytoskeletal regulations. The signal from Wnt and its receptor Fz is relayed through Dsv to LRRFIP1/GCF2, which transduces the signal to two pathways. One pathway, the β-catenin dependent Wnt/canonical pathway, leads to the nucleus to activate the transcription of genes, such as c-myc and cyclin D1, and hormone responsible genes. The other pathway, the β−catenin independent Wnt/noncanonical pathway, leads to RhoA, which regulates the cytoskeletal system through LARG. The signals from LPA and GPCR, and from extracellular matrix, such as fibronectin, also lead to RhoA through LARG. The signal that directs LRRFIP1/GCF2 to repress gene transcription was not revealed and is depicted as a question mark (?) in this figure. It is likely, however, that the transcriptional repression by LRRFIP1/GCF2 is induced by the overgrowth of cells, such as the ones treated with the growth factor [19]. Although it is not depicted in this figure, Fli-1 also binds to NRs and their coactivators to enhance the transcription of hormone responsible genes [11]. Abbreviations used; LRRFIP1/GCF2: Leucine Rich Repeat of Flightless-1 Interacting Protein 1/ GC-binding factor 2, Fz: Fizzled, Dsv: Dishevelled, LPA: Lysophosphatidic acid, GPCR: G protein coupled receptor, Fli-1: Flightless-1, LARG: Leukemia associated Rho-specific guanine nucleotide exchange factor, Trs Act: Transcriptional activator, GRIP: Glucocorticoid receptor interacting protein, CBP: Creb-binding protein, LEF1(TCF): Lymphoid enhancer-binding factor 1(T cell factor), EAAT2(GLT-1): Excitatory amino acid transporter 2 (Glutamate transporter 1), NR: Nuclear receptor.

**Figure 3 cells-08-00108-f003:**
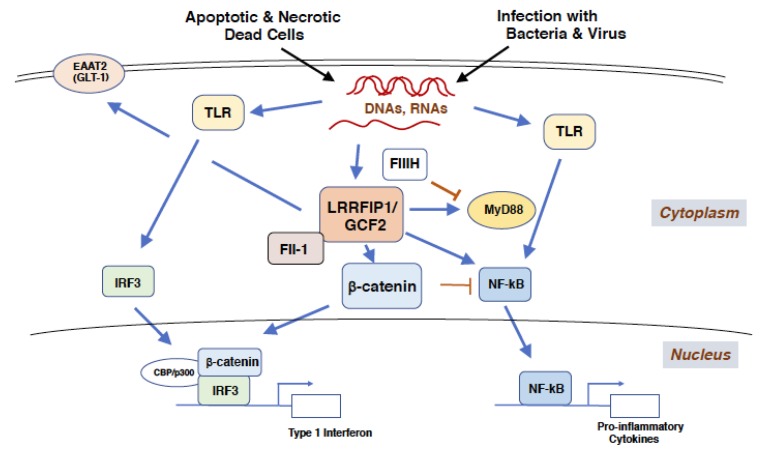
**LRRFIP1/GCF2 binds to exogenous and endogenous nucleic acids and activates type 1 interferon and pro-inflammatory cytokine responses**. Single and double stranded-nucleic acids, derived from infections with bacteria and virus and from necrotic and apoptotic dead cells, are bound to LRRFIP1/GCF2 to activate IFN-βgene transcription through β-catenin and IRF3. β-catenin is phosphorylated at Ser552 by the infections, potentially by protein kinase A or Akt [9,38]. The nucleic acids are also bound by TLRs, which activate NF-κB to produce pro-inflammatory cytokines. LRRFIP1/GCF2 competes with FliiH to interact with MyD88 so as to stimulate the TLR/NF-κB pathway. In non-virulent infection, β-catenin binds directly to and inhibits NF-κB, negatively regulating inflammation [43]. Although LRRFIP1/GCF2 represses the variant promotor of EAAT2(GLT-1) gene, LRRFIP1/GCF2 increases the level of EAAT2(GLT-1) protein in cerebral cells, which could prevent the excitotoxicity of a stroked brain. These two pathways are crucial responses to microbial infections and cell death, such as by brain ischemia. Abbreviations used; LRRFIP1/GCF2: Leucine Rich Repeat of Flightless-1 Interacting Protein 1/GC-binding factor 2, TLR: Toll like receptor, Fli-1: Flightless-1, FliiH: Flightless-1 homologue, MyD88: Myeloid differentiation factor 88, CBP: Creb-binding protein, IRF3: Interferon regulatory factor 3, EAAT2(GLT-1): Excitatory amino acid transporter 2 (Glutamate transporter 1).

**Figure 4 cells-08-00108-f004:**
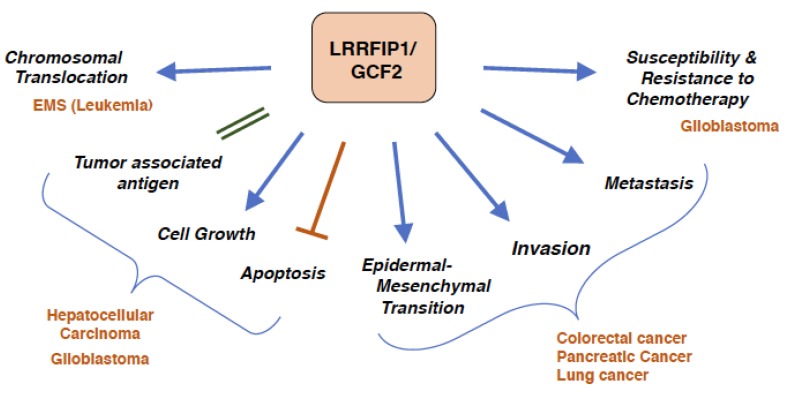
**Cancers and LRRFIP1/GCF2.** Overexpression of LRRFIP1/GCF2 plays a role in the development of cancer and malignant phenotypes, and the inhibitions of the expression reverse the malignant phenotypes. *Chromosomal Translocation*; an in-fusion transcript of LRRFIP1/GCF2 and FGFR1 gives rise to a leukemia with 8p11 myeloproliferative syndrome (EMS) [70]. *Tumor associated antigen*, *Cell Growth and Escape from Apoptosis*; LRRFIP1/GCF2 is a tumor associated antigen as it is highly expressed in hepatocellular carcinoma (HCC), and knockdown of LRRFIP1/GCF2 causes growth inhibition of HCC and glioblastoma cell lines and apoptosis of HCC cell lines [72,74]. *Epidermal Mesenchymal Transition, Invasion and Metastasis*; LRRFIP1/GCF2 promotes epithelial mesenchymal transition (EMT) in colorectal, pancreatic, and lung cancers, and silencing of LRRFIP1/GCF2 reverses EMT and inhibits migration and invasion of the cancers [15,73]. *Susceptibility and Resistance to Chemotherapy*; Resistance of a glioblastoma cell line to teniposide, a topoisomerase II inhibitor, is mediated by the down regulation of LRRFIP1/GCF2 expression. Up-regulation of LRRFIP1/GCF2 makes the glioblastoma cell line sensitive to VM-26, and glioblastoma patients with higher LRRFIP1/GCF2 expression have better prognosis with teniposide treatment [75]. The epidermoid carcinoma cell lines with over-expressed LRRFIP1/GCF2 were reported to be sensitive to doxorubicin and vincristine, but resistant to cisplatin, and cisplatin-resistant cell lines expressed higher level of LRRFIP1/GCF2 [29].

**Table 1 cells-08-00108-t001:** LRRFIP1/GCF2 and Human Diseases. †.

System/Process	Functional Roles of LRRFIP1/GCF2	Clinico-Pathological States When Dysregulated	Ref ‡
Immune	Binding to nucleic acids and induction of type 1 interferon through TLR and β-catenin pathways.	Vulnerability to Infection. §	[9,41]
	Regulation of TNF-α gene transcription	Autoimmune Diseases.	[21,44]
Neurological	Regulation of EAAT2 gene transcription and its protein level.	Excitotoxicity after Stroke	[22]
	Regulation of signal transduction pathways involving β-catenin and TLR.	Potential delayed response to Ischemia of brain.	[51]
Cardiovascular	Regulation of Platelet cytoskeleton.	Targeting inhibits Thrombosis	[59]
	Stimulation of vascular smooth muscle proliferation	Targeting inhibits Neointimal Hyperplasia of injured artery	[60]
Metabolic	Regulation of adiposity through possible involvement of TLR signaling.	SNP variants are associated with Adiposity/Obesity.	[63]
Wound Healing	Stimulation of dermal cell proliferation, inhibition of TLR4-mediated inflammation and anti-scarring in wounded legions of skin.	Delay in Wound Repair. §	[66]

† Non-malignant diseases are described in this table and the roles in cancers are depicted and described in Figure 2. ‡ Reference. § Although there has been no published report on the experimental and clinical assertions made, the described states can occur.

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
