# Peer review of "Multidisciplinary Roles of LRRFIP1/GCF2 in Human Biological Systems and Diseases"

_cells, 2019, doi:10.3390/cells8020108_

Round 1
Reviewer 1 Report
The current work is technically sound and well presented. It is well structured and provide evidences for the proof of concept.
However, there are additional concerns:
1. Provide a representative figure highlighting the biological Function of LRRFIP1/GCF2 (mechanistic outlook).
2. Figure 2 needs to be amended.
Author Response
To comment #1
New figures were provided for mechanistic functions of LRRFIP1/GCF2 in
figure 2 and 3 at the ends of section 2-3 (from the line 200) and of
section 3-2 (from the line 287) of revised manuscript, respectively.
To comment #2
Figure 2 was replaced with new figure 4 in the section 3-6 (from the
line 426) in revised manuscript.
Reviewer 2 Report
This is very interesting review, that focuses not only on basic science but also indicate the potential utility of this system in clinical medicine.
I have no critical remarks, however in the line 343 "which span several biomedical" please consider to substitute word biomedical - perhaps biological and pathophysiological systems would sound better?
Author Response
The sentence in line 343 in previous manuscript “..which span
several biomedical…” was replaced with “..which span several
pathophysiological …” in the lines 458 and 459 in revised
manuscript.
Reviewer 3 Report
Reviewer report
Recommendation: Minor revisions
This manuscript by Masato Takimoto summarized research progress of LRRFIP1/GCF2 in human biological systems through the identification of gene and protein, biological function and implication in human diseases. Especially detailed the biological function of LRRFIP1/GCF2 as transcriptional repressor, regulator for cytoskeletal system and regulator in signal transductions and list the experimental and clinical pathological implication of LRRFIP1/GCF2 dysregulations, such as autoimmune diseases, excitotoxicity of stroke, thrombosis formation, inflammation and obesity, and wound healing process, and in cancers. The author have made organized, logical and legible summary and such an comprehensive, detailed review is necessary for studying LRRFIP1/GCF2 for possible physical and clinical function. In view of this, the manuscript may be accepted for publication after minor revise.
1. Introduction: “Genomic DNA size of LRRFIP1/GCF2 is about 73 kbp and consists of 24 exons” (line 51). The exon count of LRRFIP1 (GCF2) in NCBI gene bank is 29, why the same gene has different exon number? Please verify the data accurately.
2. Could you re-check and modify the messy code of figure 2, and the font format of figure 1 and figure 2 are different.
3. Table 1 is valuable and suggestive for learning background and exploring the research direction of LRRFIP1/GCF2. Generally, the roles of protein in biological system are related to the pathological effect of diseases. More discussion about the relationship of role of LRRFIP/GCF2 in biological system/process and human diseases are recommended, such as the author mentioned “Bindings of LRRFIP1/GCF2 to nucleic acids derived from pathogens triggers an innate immune response through -catenin dependent and Toll-like receptor (TLR) pathways” (line 145-147) in section 2-3. according to previous reports, Toll-like receptor (TLR) agonists treatment could diminishes the inflammatory response to stroke and at the same time enhances the production of anti-inflammatory cytokines and neuroprotective mediators. So the LRRFIP1/GCF2 might associate with immune and neurological system though TLR pathway. The author didn’t show up the deep relationship of these biological roles and human diseases.
4. References: the format of references 8 and 46 (pages) requires to be consistent with others.
Author Response
To comment #1
The number of exons was corrected as 29 in the line 50 in
revised manuscript.
To comment #2
Figure 2 in previous manuscript was revised as new figure 4
in the section 3-6 (from the line 426) in revised manuscript.
The font Helvetica was used in all through the figures and
Calibri in all through figure legends in revised manuscript.
To comment #3
In response to this reviewer’s comment, the most parts of
section 2-3 and 3-2 were rewritten to include more
references and discussion.
The sentence that refers to ischemic tolerance, in which
pretreatments with TLR agonist diminish the inflammatory
response to stroke and protect brain, was included in the
line 267 - 269 of revised manuscript. The association
between immune and neurological system was described in
the line 269 – 270.
To comment #4
As reference list was made automatically by an application
software, EndNote, all the format for references is consistent
in revised manuscript.